# Terrestrial Species of *Drouetiella* (Cyanobacteria, Oculatellaceae) from the Russian Arctic and Subarctic Regions and Description of *Drouetiella ramosa* sp. nov.

**Denis Davydov** [1,*] , **Anna Vilnet** [1] , **Irina Novakovskaya** [2] **and Elena Patova** [2]

1   Polar-Alpine Botanic Garden-Institute of Kola Science Center of the Russian Academy of Sciences, Botanical Garden, 184256 Kirovsk, Russia
2   Institute of Biology of Komi Science Centre of the Ural Branch of the Russian Academy of Sciences, Kommunisticheskaya, 28, 167000 Syktyvkar, Russia
*   Correspondence: d.davydov@ksc.ru

**Abstract:** The strains of *Drouetiella* species (Cyanobacteria, Oculatellaceae) from a terrestrial biotope were isolated and characterized using an integrative approach including molecular, morphological, and ecological information. The specimens were collected from the Arctic and Subarctic areas of European Russia. *Drouetiella* species possess morphological plasticity and can be confused with similar species of Oculatellaceae or Leptolyngbyaceae. The 16S rRNA gene phylogeny supported the strong monophyly of the genus *Drouetiella* with the separation of four linages corresponding to three known species and one to new taxon. The 16S-23S ITS rRNA sequences of the analyzed *Drouetiella* strains differ in length and nucleotide composition, which has had an effect on the hypothetical secondary structures of the D1–D1′, Box-B, V2, and V3 helices. As a result of complex study of the genus *Drouetiella*, a new species—*Drouetiella ramosa* sp. nov.—is described from the Subarctic of European Russia.

**Keywords:** the Arctic; molecular phylogeny; 16S rRNA; 16S-23S ITS rRNA; filamentous cyanobacteria

## 1. Introduction

Cyanobacteria are oxygenic micro-phototrophs present in all ecosystems. They are often recognized as the most important primary producers [1], especially in polar ecosystems [2]. Despite a long and rich history of cyanobacterial research in the Eurasian Arctic and Subarctic [3], the existence of diversity in these regions is still greatly underestimated. An effort to widen the knowledge of cyanobacterial diversity in high latitudes is necessary.

Several studies of cyanobacteria in Svalbard [4–9], Murmansk Region [10–13], Karelia Republic [14], Komi Republic, and Yamalo-Nenets Autonomous Okrug [15–19] suggest that polar and boreal cyanobacterial floras probably contain many undiscovered species and genera.

In recent years, the cyanobacterial phylogeny has been intensely studied. Simple filamentous cyanobacteria have undergone extensive taxonomic revision with an integrative approach that resulted in the transfer of the recently established genera *Oculatella* [20] and *Timaviella* [21] into a new family, Oculatellaceae, in which six new genera were simultaneously described: *Cartusia*, *Drouetiella*, *Kaiparowitsia*, *Komarkovaea*, *Pegethrix*, *Tildeniella* [22]. The subsequent taxonomical studies increased the number of known genera in Oculatellaceae to 15 with the description of *Thermoleptolyngbya* [23], *Elainella* [24], *Shackletoniella* [25], *Aerofilum* [26], *Trichotorquatus* [27], *Amphirytos* [28], and *Siamcapillus* [29].

Unfortunately, these new species and genera are quite difficult to identify by light microscopy due to a lack of clear morphological apomorphies.

The genus *Drouetiella*, with three known species [22], could attend to such problematic taxa. This genus comprises a thin solitary filamentous false-branching form with

colorless sheaths and untapered trichomes, slightly constricted at the cross-walls. During the distinguishing of the new genus *Drouetiella*, Mai et al. [22] described within it two new species—*Drouetiella fasciculata* Mai, Johansen and Bohunická, and *Drouetiella hepatica* Mai, Johansen and Pietrasiak—from strains cultivated in the Algal Culture Collection at John Carroll University, Cleveland, USA. The authors also transferred *Phormidium luridum* Gomont to *Drouetiella lurida* (Gomont) Mai, Johansen and Pietrasiak. That species was described as *Leptothrix lurida* Kützing [30], due to its origin in a Stuttgart waterbody.

In our previous study, concerning the genus *Phormidesmis'* systematics based on nucleotide sequence data of the 16S rRNA gene and 16S-23S ITS rRNA region [13], we additionally tested five strains that were morphologically attributed to the genus *Drouetiella*; two of them, with major doubts concerning their morphological evidence, were attended to by *D. hepatica,* and three remained undetermined. The difficulties in morphological recognition of the *Drouetiella* strains cultivated in our laboratory forced us to sample published molecular data for *Drouetiella* spp., and reference strains from the family Oculatellaceae, to clarify the level of species variability and correlate this with morphological features.

This study aimed to determine the diversity of the genus *Drouetiella* from the Arctic and Subarctic regions using a combined approach: morphological analysis, and phylogenetic analysis, based on DNA sequence, genetic variability of the ribosomal gene, and secondary structures of the 16S-23S ITS rRNA region, as well as by ecological characterization.

## 2. Materials and Methods
### 2.1. Sampling

Five cyanobacterial samples were collected during the summers of 2009–2019 from different locations in the terrestrial habitats of the Russian Arctic and Subarctic area (Table 1, Figure 1).

**Table 1.** The description of the sampling sites.

| Number of Location | Number of the Strain in Collection (GB Accession Number) | Locality | Habitats | Latitude N | Longitude E | Elev. (m) |
|---|---|---|---|---|---|---|
| 1 | KPABG 4163 (ON897679) | Yamalo-Nenets Autonomous Okrug. The Polar Urals Mountains. The Ochetyvis Valley. The left shore of the Ochetyvis River. | On the limestone rocks and boulders underwater. | 68.18995 | 65.67754 | 167 |
| 2 | KPABG 132178 (ON897678) | Yamalo-Nenets Autonomous Okrug. The Polar Urals Mountains. The Ochetyvis Valley. The left shore of the Ochetyvis River. | On a wet limestone wall of rock, near the water. | 68.19059 | 65.67815 | 154 |
| 3 | KPABG 41662 (ON897680) | Murmansk Region. Ash dumps of the Apatitsky Thermal Power Plant. | Anthropogenic habitat, crust on the sludge. | 67.599972 | 33.48128 | 202 |
| 4 | SYKOA C-013-09 (ON897677) | Komi Republic. Subpolar Urals Mountains. | Next to a quartz mine on damp quartz sand. | 65.22639 | 60.24528 | 850 |
| 5 | KPABG 610005/SYKOA C-002-10 (ON897681) | Komi Republic. Subpolar Urals Mountains. Near Maloe Balbanty Lake. | Grass–moss community next to a deer camp, in the soil. | 65.155833 | 60.231833 | 690 |

Two samples (Figure 1, plots 1, 2) were collected in the northern part of the Polar Urals Mountains in the tundra zone, which is primarily composed of tundra shrubs and moss–lichen communities, by D. Davydov. The samples were found in mats with other cyanobacteria.

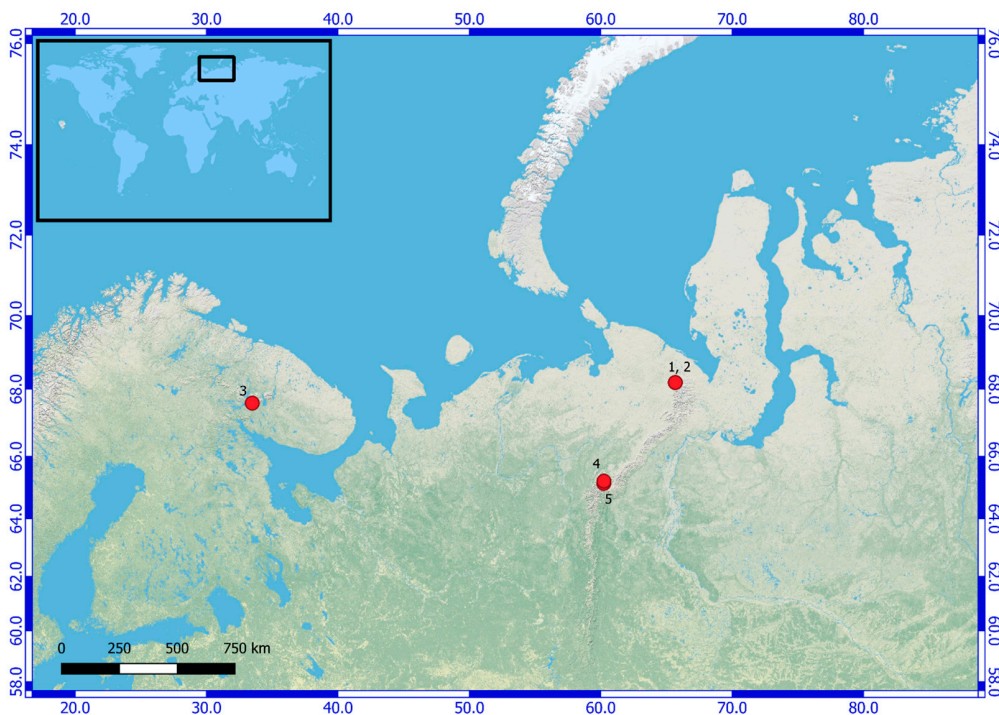

**Figure 1.** The position of the sample plots of *Drouetiella* strains; numbers of sample plots as outlined in Table 1.

Two samples (Figure 1, plots 4, 5) were collected in the Subpolar Urals Mountains by E. Patova and I. Novakovskaya, next to a quartz mine on damp quartz sand (plot 4), and in a grass–moss community next to a deer camp, in the soil (plot 5).

The single specimen (Figure 1, plot 3) was obtained from the crust on the sludge of ash dumps at the Apatitsky Thermal Power Plant, by D. Davydov.

The soil samples were collected within a 3–5 cm$^2$ area and 2 cm deep. The cyanobacterial samples were taken and dried in sterile paper bags and then air-dried and stored in the laboratory before enrichment cultivation [31].

### 2.2. Isolation of Strains

In the laboratory, the cyanobacterial strains were isolated from mixed cultures (the samples from plots 1–3 were cultivated on liquid Z8 medium [32,33], and the samples from plots 4–5 were cultivated on Agar medium with soil extract). Unialgal cultures were obtained by picking material from the edge of discrete colonies that had been growing for three weeks on solid BG11 media [33]. Cultures were maintained under artificial illumination, 16 h light 35 μmol photons m$^{-2}$s$^{-1}$/8 h dark photoperiod, at 22 °C.

The strains were deposited into the Collection of Cyanoprokaryotes at the Polar-Alpine Botanic Garden-Institute (KPABG), Apatity, Russia. Two of them were stored in the collection of microalgae at the Institute of Biology of Komi Scientific Centre (SYKOA), Syktyvkar, Russia. A portion of the growing material was dried and deposited in the herbarium at KPABG. The label information was included in the "L." information system [34].

### 2.3. Morphological Characterization

The morphological characters of the strains were described from unialgal cultures using a Zeiss AxioScope A1 microscope (Jena, Germany) equipped with Nomarski DIC optics and an Olympus DP23 camera (Tokyo, Japan). Morphometric measurements were taken using an Olympus cellsSens Entry 3.2 (Tokyo, Japan). The diacritical morphological traits used in the species' descriptions were considered, including the width and length of the cells, shape of cells, presence or absence of constrictions at the cross-wall, presence of necridic cells, color of the sheath, and the presence or absence of false-branching.

### 2.4. Specimens Sampling

The sequences of the 16S rRNA gene and the single operon of the 16S-23S ITS rRNA region for the five strains of *Drouetiella* cultivated in KPABG were obtained during our previous study, using the protocols described there [13]. These formed the base of the current estimation together with molecular data from five *Drouetiella* strains downloaded from GenBank [35]; this was also carried out for 46 of the predominantly referenced strains of allied taxa from Oculatellaceae and the outgroup. The GenBank accession numbers for the *Drouetiella* strains from KPABG are given in Table 1.

### 2.5. Molecular Analyses

Two datasets were produced for phylogenetic estimation in BioEdit 7.0.1. [36]. The first dataset included the sequences of the 16S rRNA gene for 56 accessions with *Gloeobacter kilaueensis* JS1 (NR121745) as an outgroup. For the genus *Drouetiella,* the alignment of the 16S-23S ITS rRNA region for 10 strains was also constructed; *Pegethrix olivacea* GSE-PSE-MK46-15A was chosen as an outgroup. The length of the 16S rRNA gene alignment was 1181 sites; and for the 16S-23S ITS rRNA region: 619 sites.

Molecular phylogenetic estimations were implemented through the maximum likelihood (ML) with IQ-TREE [37] and the Bayesian approach with MrBayes v. 3.2.1 [38]. The ML analysis of the 16S rRNA gene included a search for the best-fit evolutionary model of nucleotide substitutions using the incorporated option, ModelFinder [39], and ultrafast bootstrapping [40] with 1000 replicates. The K2P+I+G was selected as the best-fit evolutionary model with four rate categories of gamma distribution to evaluate the rate of heterogeneity among the sites. For the ML analysis of the 16S-23S ITS rRNA region, the model TPM3u+F+G4 was selected, and 1000 replicates for ultrafast bootstrapping were implemented.

The Bayesian analysis was only conducted for the 16S rRNA gene, using the GTR+I+G model and gamma distributions with four rate categories, as recommended by the program creators. Two independent runs of the Metropolis-coupled MCMC were used to sample the parameter values in proportion to their posterior probability. Each run included three heated chains and one unheated chain, and two starting trees were chosen randomly. Chains were run for one million generations and trees were sampled every 100th generation. The software tool Tracer [41] revealed an effective sample size of 1408.4793 and an autocorrelation time of 1278.116. The 10,000 were obtained in each run, and the first 2500 trees were discarded as burn-in. Thereafter, 15,000 trees were sampled from both runs. The average standard deviation of the split frequencies between the two runs was 0.007279. Bayesian posterior probabilities in both estimations were calculated from trees sampled after burn-in for each run as well. The majority rule (MJ) consensus tree for both datasets was calculated after combining the runs minus a burn-in of 25%.

The infrageneric and infraspecific similarity of the 16S rRNA gene and 16S-23S ITS rRNA region of the tested *Drouetiella* strains were calculated with the following formula $100 \times (1 - p)$, where p is as the average pairwise *p*-distances inferred from Mega 11 [42].

The hypothetical secondary structures for the four conserved domains of the 16S-23S ITS rRNA region were determined using the program Mfold Ver. 3.1 [43] and prepared for publication using Adobe CorelDRAW 24.0.0.301.

### 3. Results

### 3.1. Morphology

All of the strains analyzed in this paper differ from the original species' descriptions [22] concerning morphological features (Table 2).

**Table 2.** Morphological comparison among the named *Drouetiella* strains. Information for the type strains were obtained from [22].

| Characteristics/ Species | D. lurida | | | | D. hepatica | | D. ramosa | D. fasciculata |
|---|---|---|---|---|---|---|---|---|
| | Lukesova 1986/6 | KPABG 4163 | KPABG 41662 | KPABG 610005 | Uher 2000/2452 | KPABG 132178 | SYKOA C-013-09 | GSE-PSE-MK29-07A |
| False-branching | - | + | + | - | very rare | - | common | - |
| Cells elongated | + | + | + | + | + | + | - | + |
| Constricted at cross-walls | not or slightly | distinctly | slightly | + | not or slightly | not or slightly | not or slightly | not |
| Necridia | - | + | + | - | frequent | + | + | + |
| Width of cells | 1.7–2.1 | 1.8–2.5 | 1.6–2.4 | 1.6–3.4 | 1.5–3.0 | 1.6–3.1 | 2.3–4.3 | 1.5–2.4 (3.0) |
| Length of cells | (2.1) 2.9–3.8 (5.4) | 1.2–2.6 | 1.9–3 | 1.5–3.8(4) | (2.2) 3.1–4.5 | 2.0–4.0 | 1.9–2.6 | 3.1–4.4 (5.4) |
| Meristematic zones | - | - | - | - | + | - | - | - |
| Hormogonia | - | - | - | rare | - | - | rare | rare |
| Coloration | liver-brown | olive-green | olive-green | blue-green, olive-green | brownish | olive-green | olive-green, blue-green | blue-green |
| Locality/ habitat | Czech Republic, temperate forest/aerophytic, soil | The Russian Arctic/epilithic on a boulder in a river, underwater | The Russian Subarctic/aerophytic, crust on the sludge | The Russian Subarctic/aerophytic, soil crusts | Slovakia, temperate forest/on subaerial seep wall and waterfall | The Russian Arctic/seep wall | The Russian Subarctic/aerophytic, soil | The USA, semi-arid/seep wall and waterfall |

The strain KPABG 4163 fits well within the dimensions of *Drouetiella lurida* species. Macroscopically, mats of *Drouetiella lurida* KPABG 4163 are green-colored on an Agar plate. Trichomes with long or isodiametric cells (1.2–2.6 μm long) are distinctly constricted at the cross-walls (Figure 2a,b). The trichomes are characterized by rare false-branching, occasionally with necridic cells. Thylakoids: parietal. Cell: olive-green.

The strain KPABG 41662 is similar to the strain KPABG 4163 (Figure 2c,d). It is characterized by long, straight trichomes with elongated or isodiametric cells (1.9–3 μm long, 1.6–2.4 μm wide) and rare false-branching.

The investigated strain of *Drouetiella lurida* KPABG 610005 is characterized by long, curved or slightly waved filaments, containing one trichome, with or without sheath, 1.6–3 (3.4) μm wide (Figure 2e,f). Mucilaginous sheaths are colorless, hyaline to firm, in older stages mostly firm. Trichomes indistinctly constricted at the cross-walls up to clearly constricted. Cells green or blue-green, from isodiametric to elongate, 1.5–3.8(4) μm long. Necridia cells absent.

The principal morphological traits, such as the size and shape of cells, and presence of necridia cells, of the strain *Drouetiella hepatica* KPABG 132178 (Figure 2g) were very similar to the reference strain *D. hepatica* Uher 2000/2452 reported by Mai et al. [22]. Minor differences in coloration and meristematic zone exist.

The strain SYKOA C-013-09 is distinguished by very common false-branching trichomes (Figure 3). Filaments: long and curved, containing one trichome. Sheath: firm, colorless, thin, usually visible only during or following hormogonia formation and at the ends of filaments. Trichomes: isopolar, untapered, not or indistinctly constricted at the cross-walls, 2.3–4.3 μm wide. Cells: isodiametric or slightly longer or shorter than wide, olive-green or blue-green, 1.9–2.6 μm long. Apical cells: rounded, the same size as regular cells. Necridia: frequent.

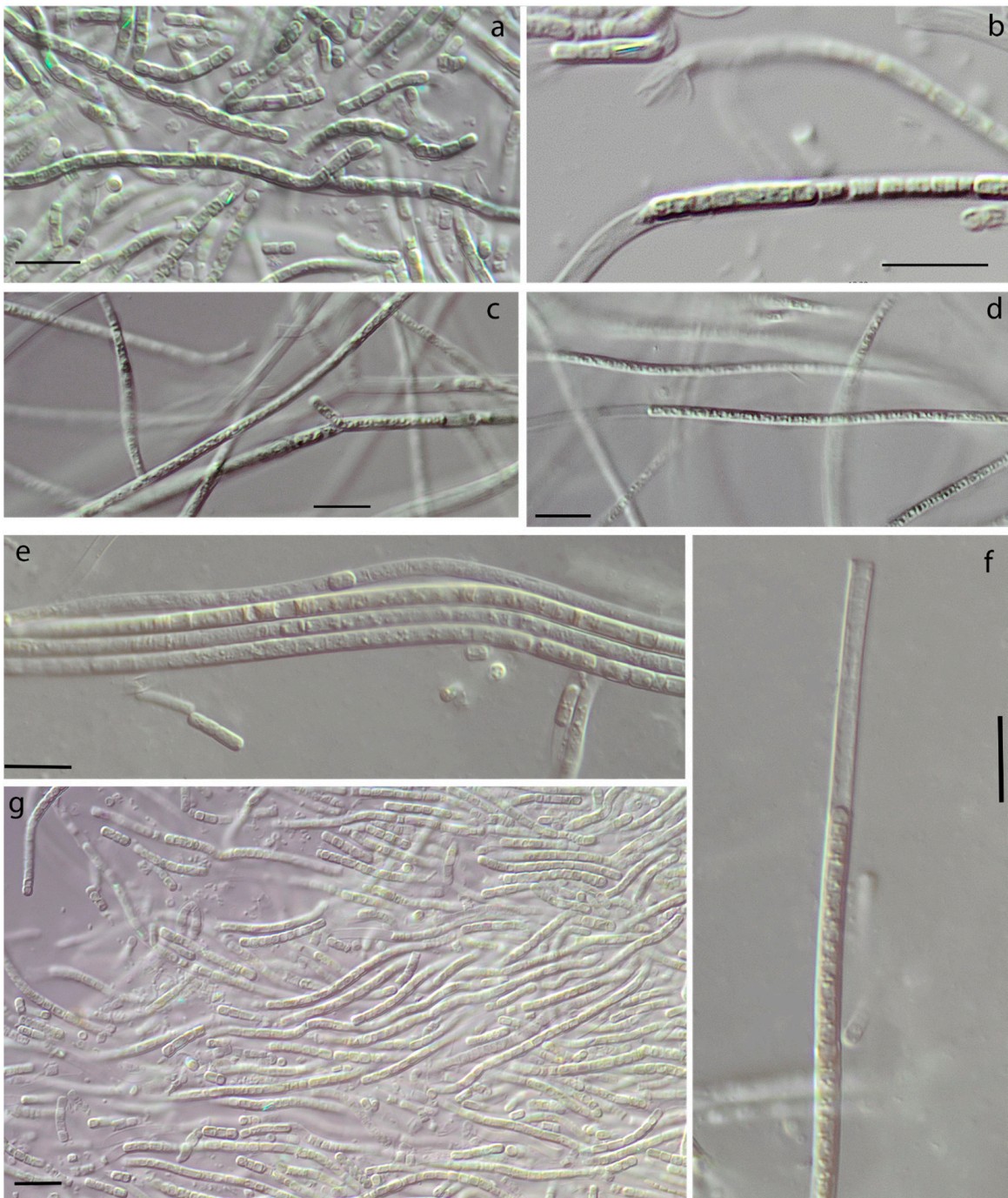

**Figure 2.** Light micrographs of *Drouetiella* strains: (**a**,**b**) *Drouetiella lurida* KPABG 4163; (**c**,**d**) *Drouetiella lurida* KPABG 41662; (**e**,**f**) *Drouetiella lurida* KPABG 610005; (**g**) *Drouetiella hepatica* KPABG 132178; (**b**,**d**,**f**) unbranching filaments with thin, firm sheaths; (**a**,**c**) false-branching filaments. All photos are at the same magnification, scale = 10 μm.

There was considerable overlap in the size ranges for all measured features (Table 2), and clear morphological separation of these taxa was not possible.

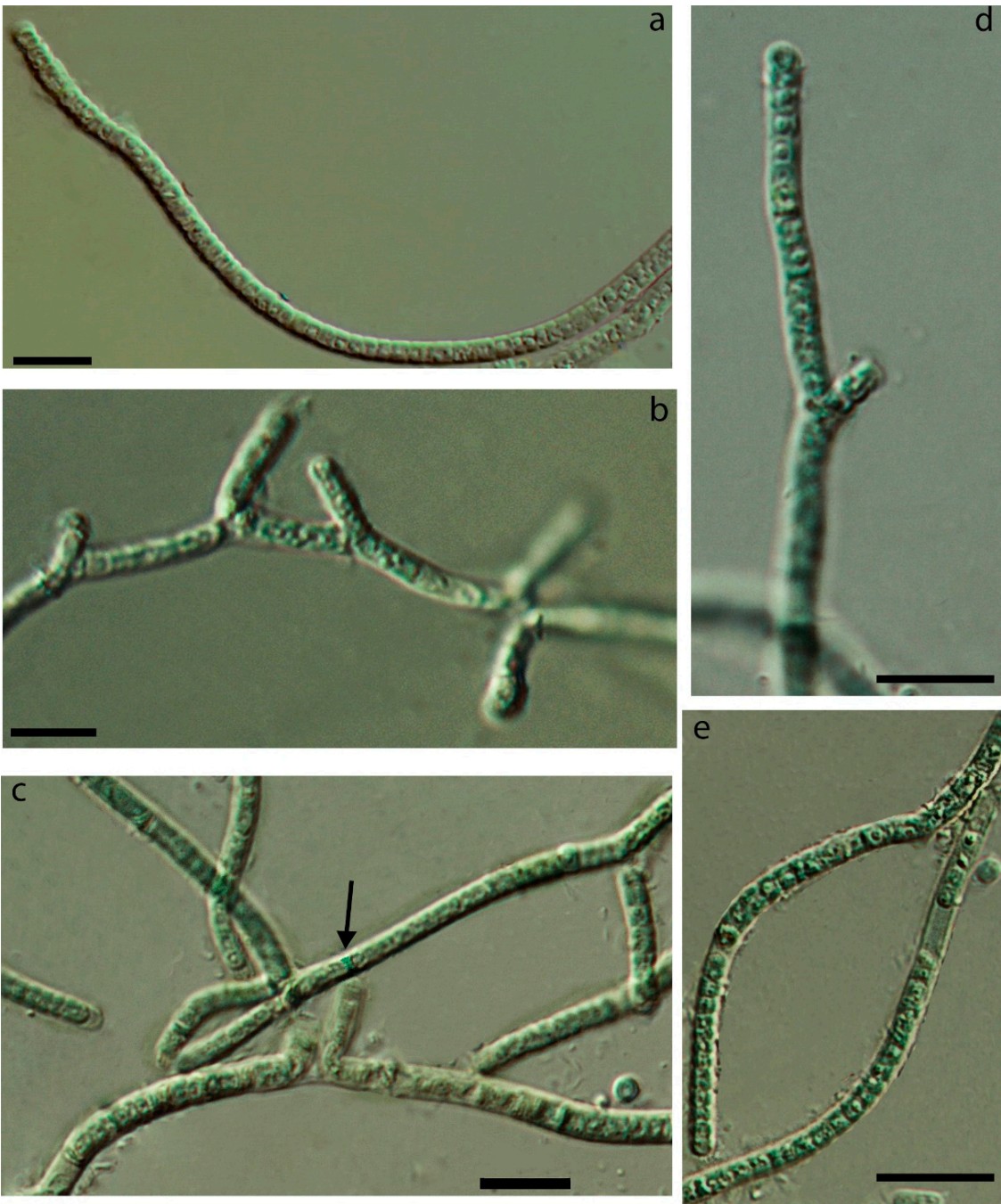

**Figure 3.** Light micrographs of the *Drouetiella* SYKOA C-013-09 strain: (**a**,**e**) unbranching filaments with thin, firm sheaths; (**b**–**d**) false-branching filaments; (**e**) filaments with thin, firm sheaths. The arrow indicates a necridia. All photos are at the same magnification, scale = 10 μm.

*3.2. Phylogeny*

The ML analysis of the 16S rRNA gene dataset resulted in a single tree with an arithmetic mean of Log likelihood of −7928.8266; in Bayesian analysis, the arithmetic means of Log likelihoods for each sampling run were −7957.81 and −7955.56. Trees from both phylogenetic estimations revealed similar topologies; thus, Figure 4 demonstrates the ML topology with an indication of the bootstrap support values (BS) and Bayesian posterior probabilities (PP). The ML analysis of the 16S-23S ITS rRNA region resulted in a single tree with an arithmetic mean of Log likelihood of −2343.4528; the obtained topology is shown in Figure 5.

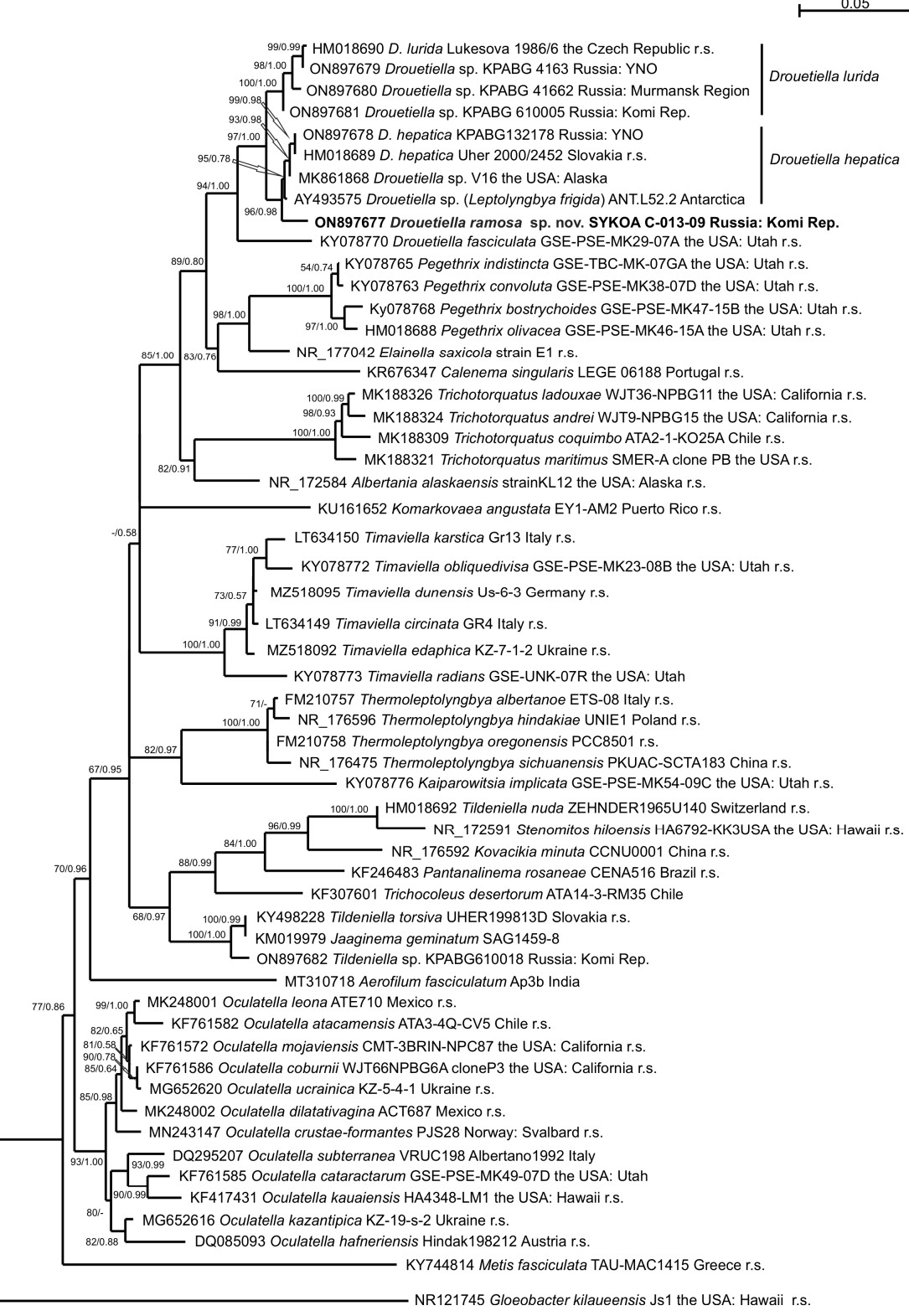

**Figure 4.** Phylogram obtained under the maximum likelihood approach for 56 accessions of Oculatellaceae based on the 16S rRNA gene. Bootstrap support values from the maximum likelihood and Bayesian posterior probabilities more than 50% (0.50) are indicated. The reference strain marked as r.s.; YNO: Yamalo-Nenets A.O.

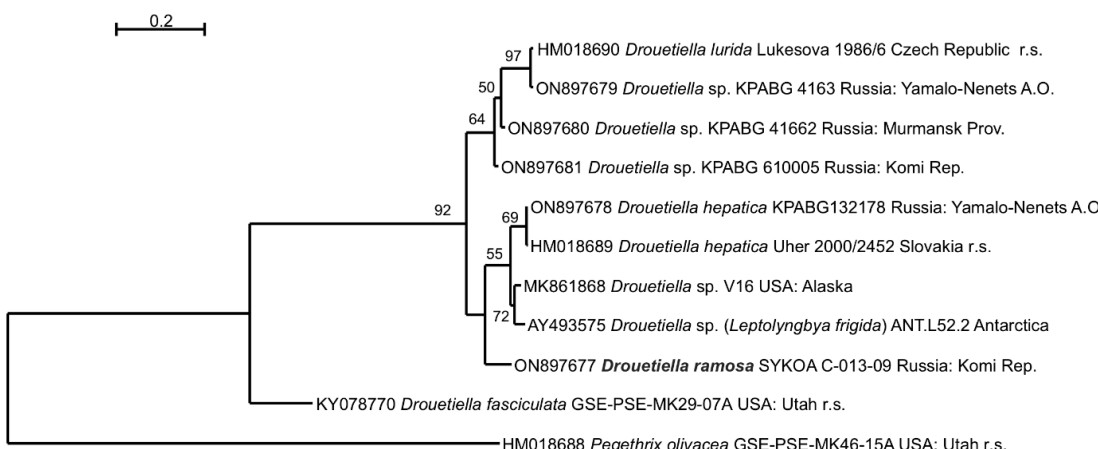

**Figure 5.** Phylogram obtained under the maximum likelihood approach for 10 accessions of *Drouetiella* based on the 16S-23S ITS region. Bootstrap support values of more than 50% are indicated.

The backbone phylogeny of the family Oculatellaceae, in the main course, agreed with that obtained previously and discussed in detail in the article [22]. Here, we focus only on affinities within the genus *Drouetiella.*

Ten strains of the genus *Drouetiella* formed a terminal clade with support BS = 94% and PP = 1.00 (or 94/1.00) in the reconstructed topology. The basal position in the clade belongs to the strain of *Drouetiella fasciculata* GSE-PSE-MK29-07A from the USA. Three unidentified Russian strains were found in the subclade (100/1.00) with the reference strain of *Drouetiella lurida* Lukesova 1986/6 from the Czech Republic. The closest relative to the reference strain was *Drouetiella* sp. KPABG 4163 from Yamalo-Nenets A.O. (99/0.99), the second divergence was presented by *Drouetiella* sp. KPABG 41662 from Murmansk Region (98/1.00), and the third by *Drouetiella* sp. KPABG 610005 from Komi Republic (100/1.00). The other Russian strain from Yamalo-Nenets A.O. revealed a sister affinity to the reference strain of *Drouetiella hepatica* Uher 2000/2452 from Slovakia (99/0.98). A subsequent relation to this subclade was composed of strains of *Drouetiella* sp. V16 from the USA (93/0.98) and *Drouetiella* sp. (*Leptolyngbya frigida*) ANT.L52.2 from Antarctica (95/0.78). The strain initially marked as *Drouetiella* sp. SYKOA C-013-09 from Komi Republic was found to have a sister relation to *D. hepatica*-subclade (96/0.98), with a comparatively long branch that supposed its separation.

The topology arising from the ML analysis of the 16S-23S ITS rRNA region was highly similar to the topology obtained for the 16S rRNA gene in branching; however, the nodes received slighter bootstrap supports (Figure 5). The strains V16 and ANT.L52.2 revealed a sister relationship, supported by a BS = 72%. The position of strain SYKOA C-013-09 was not supported.

The variability of the sequence similarity values within the subclade containing the reference strain of *Drouetiella lurida* Lukesova 1986/6 was consistent in 99.17–99.90% cases in the 16S rRNA gene and 93.53–99.12% in the 16S-23S ITS rRNA region (Table 3).

The strain KPABG 4163 was highly similar to the reference strain of the type species; strains KPABG 41662 and KPABG 610005 were similar to each other. The level of similarity for both strain pairs was 99.17–99.64% in the 16S rRNA gene and 93.53–94.83% in the 16S-23S ITS rRNA region. The similarity within the subclade containing the reference strain of *Drouetiella hepatica* Uher 2000/2452 was higher and achieved 99.50–100% in the 16S rRNA and 94.15–99.40% in the 16S-23S ITS rRNA region. Yarza et al. [44], pointed out that a similarity of more than 98.7% in the 16S rRNA gene allows us to consider allied strains as members of a single cyanobacterial species. Gonzalez-Resendiz et al. [45] suggested that less than 3% difference in ITS agrees with a level of infraspecific variability, and a difference of more than 7% determines a different species. In terms of *Droutiella* study, the level of 16S rRNA gene similarity fits well in a species concept, whereas in the 16S-23S ITS rRNA region,

similarity takes an intermediate position within 3–7%. According to the trees' topology, the tested strains from Russia could be attributed to the corresponding species *Drouetiella lurida* (KPABG 4163, 41662, 610005) and *D. hepatica* (KPABG 132178). The sequence similarity of both species was 97.81–98.75% in the 16S rRNA and 85.14–89.10% in the 16S-23S ITS rRNA locus. The strain *Drouetiella* sp. V16 was highly similar in its 16S rRNA gene sequence with *Drouetiella hepatica* Uher 2000/2452 (99.80%) and *Drouetiella hepatica* KPABG 132178 (99.82%), and in its 16S-23S ITS rRNA, with a 95.57% similarity for both strains. The strain *Drouetiella* sp. ANT.L52.2 had $\geq$ 99.55% identity in the 16S rRNA gene and $\geq$94.15% in the 16S-23S ITS rRNA, compared with other strains of the *Drouetiella hepatica* clade. Evidently, the level of similarity of both previously sequenced strains suggests them as belonging to *Drouetiella hepatica*.

**Table 3.** The nucleotide sequence similarity for the genus *Drouetiella* strains, based on the nucleotide sequence data of the 16S rRNA gene and 16S-23S ITS rRNA region, %.

| Taxon | Nucleotide Sequence Similarity, 16S/16S-23S ITS, % | | | | | | | | |
|---|---|---|---|---|---|---|---|---|---|
| | **1** | **2** | **3** | **4** | **5** | **6** | **7** | **8** | **9** |
| 1. *D. lurida* Lukesova 1986/6 | - | | | | | | | | |
| 2. *D. lurida* KPABG 4163 | 99.90/ 99.12 | - | | | | | | | |
| 3. *D. lurida* KPABG 41662 | 99.48/ 94.83 | 99.64/ 94.13 | - | | | | | | |
| 4. *D. lurida* KPABG 610005 | 99.17/ 93.53 | 99.37/ 93.72 | 99.28/ 97.42 | - | | | | | |
| 5. *D. hepatica* KPABG 132178 | 97.82/ 85.58 | 98.21/ 85.22 | 98.40/ 87.60 | 98.66/ 88.15 | - | | | | |
| 6. *D. hepatica* Uher 2000/2452 | 97.81/ 85.58 | 97.93/ 85.14 | 98.14/ 87.88 | 98.45/ 88.41 | 100.00/ 99.40 | - | | | |
| 7. *Drouetiella* sp. V16 | 97.81/ 87.08 | 98.21/ 87.32 | 98.40/ 88.94 | 98.66/ 89.10 | 99.82/ 95.57 | 99.80/ 95.57 | - | | |
| 8. *Drouetiella* sp. ANTL522 | 97.91/ 86.67 | 98.30/ 86.35 | 98.48/ 87.53 | 98.75/ 87.45 | 99.55/ 94.15 | 99.50/ 94.22 | 99.74/ 96.98 | - | |
| 9. *D. ramosa* SYKOA C-013-09 | 98.02/ 88.99 | 98.39/ 87.73 | 98.03/ 88.10 | 98.30/ 88.89 | 98.84/ 89.41 | 98.65/ 89.24 | 98.84/ 91.57 | 98.92/ 90.18 | - |
| 10. *D. fasciculata* GSE-PSE-MK29-07A | 96.42/ 79.57 | 96.51/ 78.57 | 96.17/ 80.14 | 98.43/ 79.91 | 96.52/ 80.59 | 96.54/ 80.59 | 96.82/ 83.33 | 96.90/ 80.41 | 96.60/ 80.14 |

The strain *Drouetiella* sp. SYKOA C-013-09 was similar to both of the mentioned *Drouetiella* species with 98.02–98.92% in the 16S rRNA gene and 87.73–91.57% in the 16S-23S ITS rRNA region. *Drouetiella fasciculata* appears to be the most isolated species in the genus that revealed a similarity with the rest of the taxa, with 96.17–98.43% in the 16S rRNA gene and 78.57–83.33% in the 16S-23S ITS rRNA region. The comparatively low level of sequence similarity in strain SYKOA C-013-09 compared to *Drouetiella lurida* and *D. hepatica*, additionally suggests its separation from known species.

### 3.3. The Secondary Structure of Conserved Domains of the 16S-23S ITS rRNA

The sequence length variation in the conserved domains of 16S-23S ITS rRNA is quite different within the genus *Drouetiella*. The length of the V3 helix was stable for all strains of *Drouetiella* and consisted of 50 base pairs (b.p.). The length of the D1-D1′ helix was only 65 b.p. for *D. fasciculata*, whereas for the other tested strains, it was 64 b.p. The sequence length variability was slightly varied in the Box-B region: with 33 b.p. counted for the strains *D. lurida* Lukesova 1986/6, and KPABG 4163, and 32 b.p. for the strains

KPABG 41662 and KPABG 610005; in addition, there were 34 b.p. for all strains allied to *D. hepatica* Uher 2000/2452 and SYKOA C-013-09; and 39 b.p. for *D. fasciculata* GSE-PSE-MK29-07A.

The greatest sequence length variability was found in the V2 region. Among the strains attended to by *Drouetiella lurida,* the length of the helices changed twice: from 16 b.p. in *D. lurida* Lukesova 1986/6 and KPABG 4163, to 25 b.p. in strain KPABG 610005, and 30 b.p. in strain KPABG 41662. All strains of the *D. hepatica* affinity had 21 b.p. in the V2 helix, except for the strain SYKOA C-013-09, which had 23 b.p. Evidently, this region is absent in the *D. fasciculata* spacer.

The D1-D1′ helix possesses three loops; only strain KPABG 610005 has four loops (Figure 6). The D1-D1′ helix secondary structures in the *Drouetiella lurida* clade strains (Lukesova 1986/6, KPABG 4163, KPABG 41662) are characterized by a long stem between the first and second loops, whereas strains of *D. hepatica*, *D. fasciculata,* and SYKOA C-013-09 have only two steps in the corresponding region. There are three types of motives in the terminal loop in *D. lurida* Lukesova 1986/6, and two in *D. hepatica.*

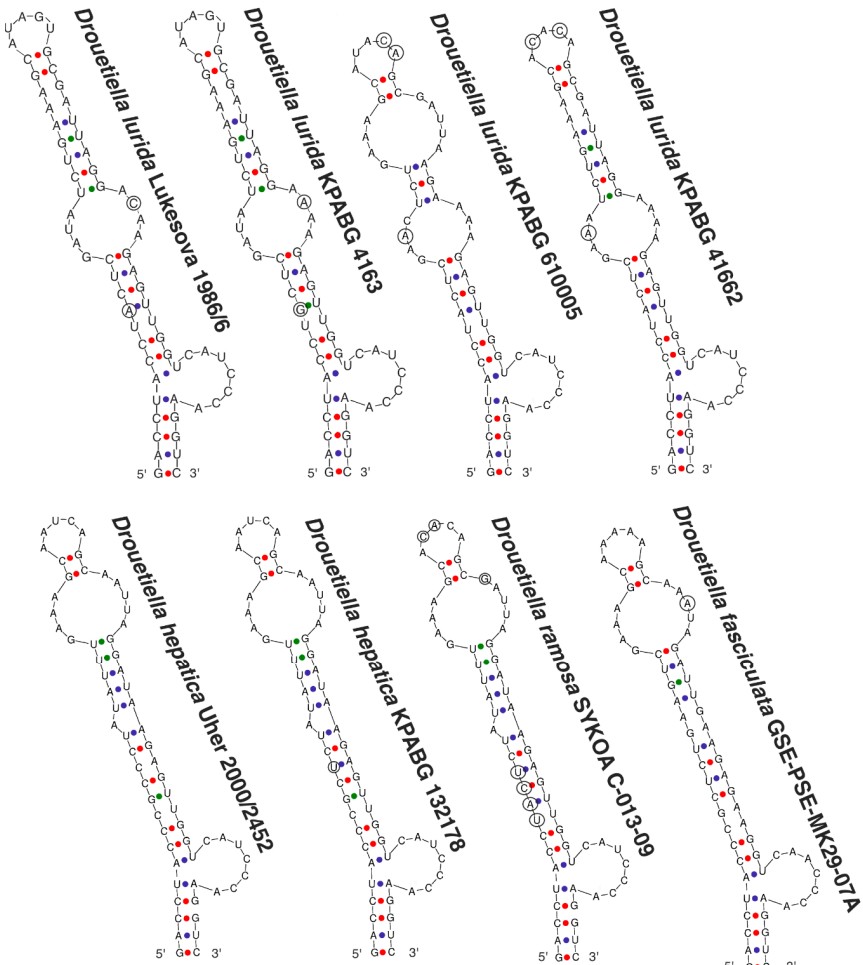

**Figure 6.** The secondary structure of D1-D1′ helices of the 16S-23S ITS rRNA region of *Drouetiella* strains. Circles indicate differences.

The secondary structures of D1-D1′ *Drouetiella lurida* Lukesova 1986/6 and *D. lurida* KPABG 4163 are highly similar: they have a substitution in the 10th position U/G and in the 42nd position C/A. The subterminal loop of *D. lurida* KPABG 610005 is similar to that of SYKOA C-013-09.

The strain *Drouetiella hepatica* KPABG 132178 has only one substitution in the 12th position C/U compared with *D. hepatica* UHER 2000/2452.

The strain SYKOA C-013-09 differs by the substitution of two nucleotides in the terminal loop, a substitution of one nucleotide in the middle loop (position 34 A/G), and substitutions of four nucleotides in the stem between the middle and basal loops (9th-12th positions) compared with the reference strain *Drouetiella hepatica* UHER 2000/2452. *Drouetiella fasciculata* has a unique motive of a terminal loop in the genus, a substitution in the middle loop compared with *D. hepatica* and SYKOA C-013-09, and a longer stem between the middle and basal loops.

The secondary structures of the V2 helices, in spite of their great length variability, possess a common view within the genus—a stem with a terminal loop (Figure 7). Three types of V2 helices were found in the *Drouetiella lurida* clade that reflected sufficient length diversity of the nucleotide sequences. The type strain of *D. lurida* and *D. lurida* KPABG 4163 has an identical structure with the shortest stem, by only six steps. The strains *D. lurida* KPABG 610005 and *D. lurida* KPABG 41662 have 9 and 12 steps in the stem, correspondingly, and different motives in the terminal loop. Four strains from the *Drouetiella hepatica* clade, with an identical length of V2 helix, have stems with nine steps, but each strain is characterized by a unique motive in the loop region presented, in each case, by only three nucleotides. The strain SYKOA C-013-09 has eight steps in a stem, a single unpaired base, and a loop composed of six nucleotides, which sufficiently differentiated this strain from multiplied sampled *Drouetiella hepatica*, with its quite stable structure. The strain *Drouetiella fasciculata* GSE-PSE-MK29-07A has the shortest region between the two tRNA genes that do not contain a V2 helix.

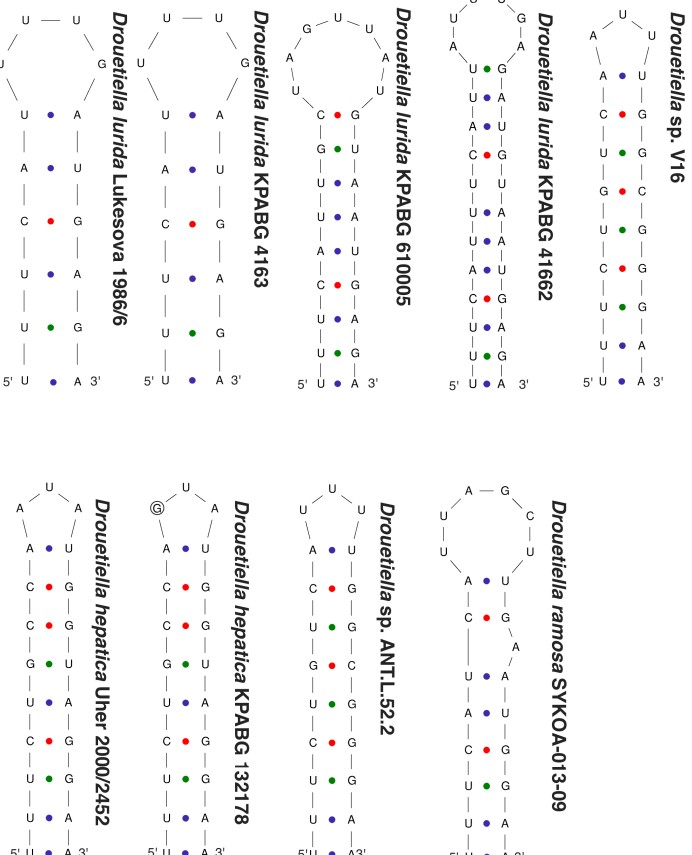

**Figure 7.** The secondary structure of the V2 helices of the 16S-23S ITS rRNA region of the *Drouetiella* strains. Circles indicate differences.

There are two types of Box-B helix structures in the genus: all strains of *D. lurida* clade are characterized by two loops; strains of the rest of the taxa only have a terminal loop (Figure 8). Three types of motives in the terminal loop are registered in the *D. lurida* clade (one type in a pair, *Drouetiella lurida* Lukesova 1986/6 and *D. lurida* KPABG 4163; the second

in a pair, *D. lurida* KPABG 610005 and *D. lurida* KPABG 41662; and the third in a strain, *Drouetiella* sp. V16). The strains of *D. hepatica* possess a single motive; the stem structures for all strains are identical. The strain SYKOA C-013-09 differs from all *D. hepatica* strains only by the single substitution G/A in the 18th position in the terminal loop. The strain *Drouetiella fasciculata* GSE-PSE-MK29-07A differs from other species by the motive of a terminal loop and a longer stem.

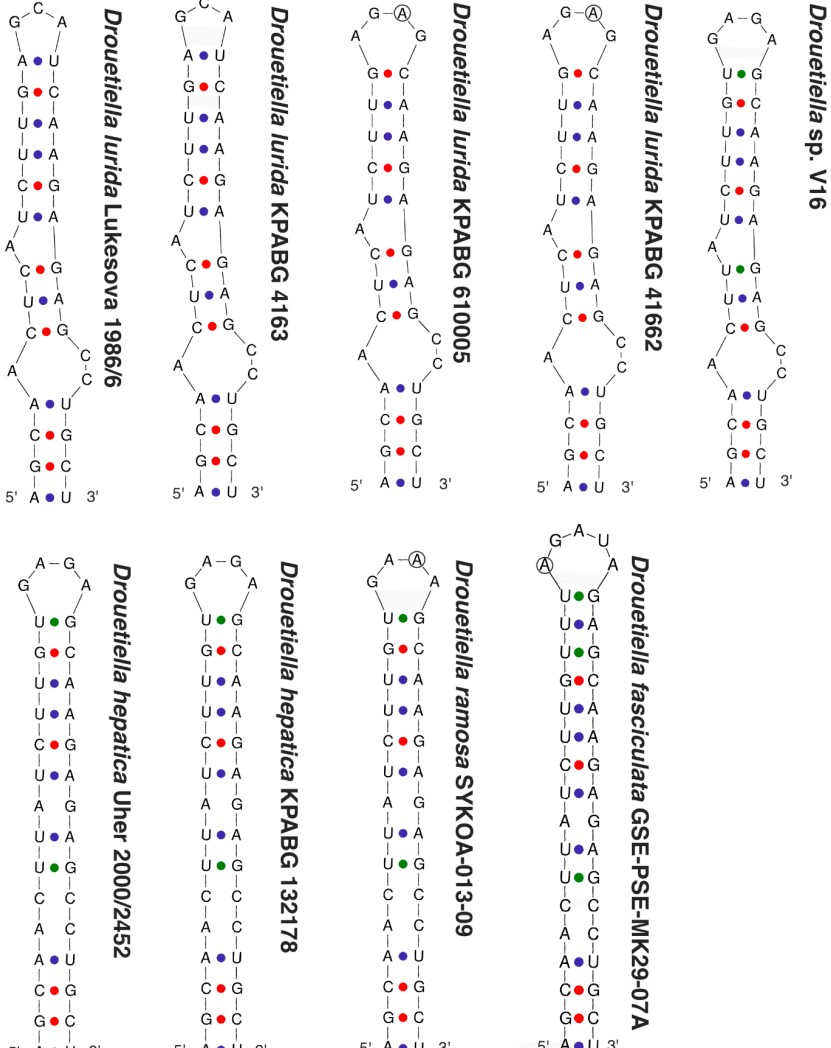

**Figure 8.** The secondary structure of the Box-B helices of the 16S-23S ITS rRNA region of the *Drouetiella* strains. Circles indicate differences.

The secondary structure of the V3 helices reveals three loops for all tested strains of the genus (Figure 9). For the type strain *Drouetiella lurida* Lukesova 1986/6, the secondary structure of the V3 region could not be reconstructed due to the absence of appropriate nucleotide sequence data. Two types of motives in the terminal loop are found for strains of *Drouetiella lurida* (the first type: *D. lurida* KPABG 4163 and *D. lurida* KPABG 610005; the second type: *D. lurida* KPABG 41662). The insertion U-A in the 15th/36th position of the stem between the terminal and middle loops is presented in strain KPABG 610005. Two substitutions—G/U in the 28th and U/C in the 31st position—are marked in strain KPABG 41662. The length of the stem between the terminal and middle loops counts 10 and 11 steps.

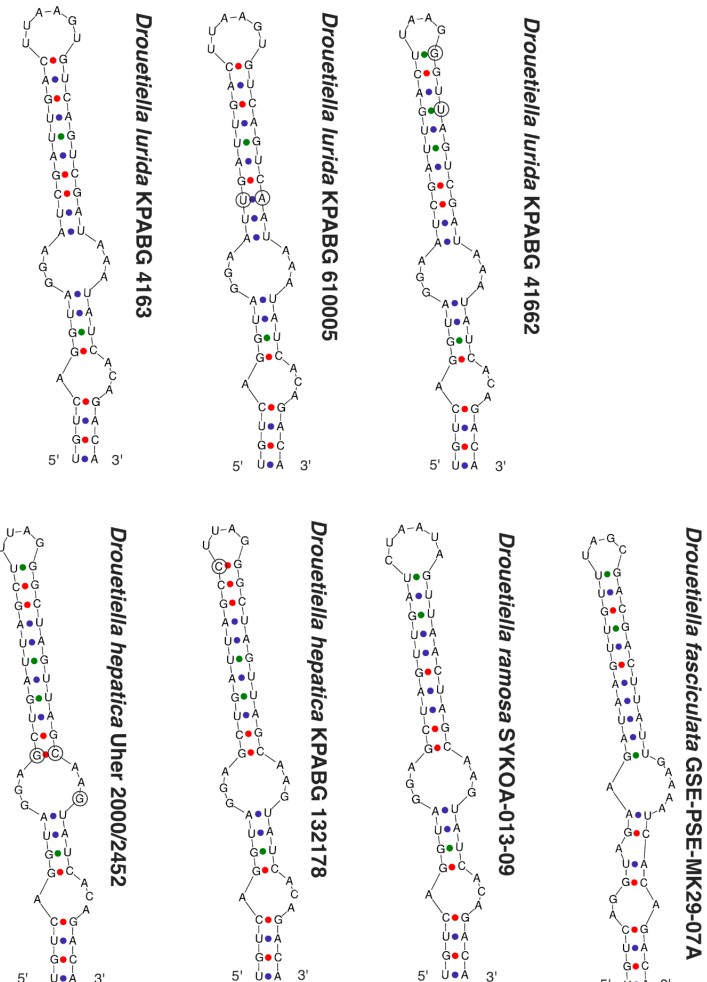

**Figure 9.** The secondary structure of the V3 helices of the 16S-23S ITS region in *Drouetiella* strains. Circles indicate differences.

Two types of motives in the terminal loop have been obtained for strains of *Drouetiella hepatica*. The strain KPABG 132178 differs from the type strain by a single substitution U/C in the base of the terminal loop.

The middle loop of the *Drouetiella hepatica* strains and SYKOA C-013-09 is common and differs from this region in *Drouetiella lurida* strains by three substitutions. The strain SYKOA C-013-09 has a unique motive of a terminal loop, and the stem between the terminal and middle loop is shorter by one step compared with strains of *Drouetiella hepatica*. The secondary structure of the V3 helix of *Drouetiella fasciculata* differs from other taxa in the motives of loops and stems.

The hypothetical secondary structures of conserved helices allow us to note significant differences between known species of the genus *Drouetiella lurida, D. hepatica*, and *D. fasciculata*, especially in terms of the D1-D1', V2, and Box-B stem loops, whereas V3 seems to be more conservative. The variability of the secondary structures in all four regions is marked for both multiplied sampled *Drouetiella lurida* and *D. hepatica.* The features of the secondary structures for strain SYKOA C-013-09 are unique and not registered fully in the sister allied species *Drouetiella hepatica,* which also allows us to segregate it taxonomically.

Taking into account the morphological features, position on the phylogenetic tree, nucleotide sequence dissimilarity, and the secondary structures of the conserved helices, we suggest that the strain SYKOA C-013-09 is a species new to science, and which will be described here.

*3.4. Taxonomic Description*

*Drouetiella ramosa* Davydov, Vilnet, Novakovskaya and Patova sp. nov. (Figure 3)

Diagnosis: *D. ramosa* is phenotypically distinct from other *Drouetiella* species due to its very common false-branching and short cells. It is different from any other representatives of the genus *Drouetiella* by its phylogenetic position based on 16S rRNA and 16S-23S ITS rRNA gene phylogenies, with differences in the secondary structures of D1-D1', Box-B, V2, and V3 helices. The V2 and V3 helices have the unique motive of a terminal loop (Figures 7 and 9). The percent dissimilarity between the 16S-23S ITS rRNA of this species and the other taxa is > 8% (Table 3).

Description: Colony green or olive-green, forming mucilaginous mats in liquid culture. Filaments long to short, curved, with or without sheaths, containing one trichome. Sheath firm, colorless, thin, usually visible only during or following hormogonia formation and at the ends of filaments. Trichomes: isopolar, untapered, not or indistinctly constricted near the transverse cell walls with very common false-branching, 2.3–4.3 μm wide. Cells: isodiametric or slightly longer or shorter than wide, olive-green or blue-green, 1.9–2.6 μm long. Apical cells: rounded, the same size as regular cells. Necridia: frequent.

Etymology: *D. ramosa* N.L. fem. Adj. = branched, bearing branches.

Holotype designated here: dried specimen deposited into the herbarium of the Polar-Alpine Botanic Garden-Institute (KPABG), Apatity, Russia, under the following accession number: KPABG 4511.

Type locality: Russia. Komi Republic. Subpolar Urals Mountains, 65.22639 N, 60.24528 E, elevation: 850 m. Next to a quartz mine on damp quartz sand, collected by E. Patova on August 5, 2009.

Reference strain: SYKOA C-013-09 (isolated into culture by I. Novakovskaya), deposited in the Culture Collection of Algae at the Institute of Biology of Komi Scientific Centre under the number SYKOA C-013-09.

NCBI GenBank Accession number: ON897677.

## 4. Discussion

Similarly to many other Oculatellaceae and Leptolyngbyaceae species, morphological convergence and a low level of plasticity complicate the morphological identification of species in the genus *Drouetiella*. A comparison of the morphological features suggested that the *Drouetiella* species showed an indistinguishable shape and structure of the trichomes and a variability in cell sizes between them; as such, we cannot distinguish them with confidence based on morphological data only. In the original description, the type species *Drouetiella lurida* is characterized by purple and blackish-violet mats [30]. Afterwards, the authors noted brown and reddish-brown colonies [22,46]. Filaments of *Drouetiella lurida* are characterized as lacking false-branching, with elongate cells of a brownish color [22].

The investigated strains KPABG 4163, KPABG 41662, and KPABG 610005 are similar to *Drouetiella lurida*, with their long and wide cells; however, our strains have trichomes characterized by false-branching (KPABG 4163, KPABG 41662), have distinctly constricted cross-walls (KPABG 4163, KPABG 610005), and have olive-green cell coloration. Such variability in color and branching could be associated with the origin of the native samples and could only be postulated by enlarged number of studied samples. The 16S rRNA and 16S-23S ITS rRNA genes phylogenies suggest a relationship between the tested strains to the type strain *Drouetiella lurida* Lukesova 1986/6. The strain KPABG 4163 is closer and more similar, regarding the secondary structure of its conservative helices in the 16S-23S ITS rRNA, to the type strain, whereas the strains KPABG 41662 and KPABG 610005 possess common features and differ from the first strains' pair. The 16S rRNA gene similarity for all tested strains was high and corresponded more closely with the level of infraspecific variation (<1.3%), whereas the similarity of the 16S-23S ITS rRNA had an interval of 3–7%, whereby it is hard to solve the problem of whether the strains belong to the same species, or to different taxa. The greatest diversity in the secondary structures of the ITS regions in the *Drouetiella lurida* clade was registered for the D1-D1' and V2 helices, whereas Box-B and

V3 appear to be more conservative. Previously, we illustrated the widely varied secondary structures in the helices among a large number of geographically remote strains of other filamentous species, *Phormidesmis nigrescens, P. priestley,* and *P. communis,* wherein the 16S rRNA gene similarity within each species was less than 1.3%, and the 16S-23S ITS rRNA was approximately 7% [13]. Evidently, additional strain sampling of *Drouetiella lurida* would allow us to estimate the species' genetic diversity and to make a robust taxonomical conclusion; however, the data in hand suggests that the attending strains, KPABG 4163, KPABG 41662, and KPABG 610005, belong to a single species, *Drouetiella lurida,* with a wide distribution stretching from East Europe to the Murmansk Region, and the Polar and Subpolar Ural Mountains.

*Drouetiella hepatica* is similar to *Drouetiella lurida* by the color of its colony and trichomes; however, this species is distinguished from the other ones by its meristematic zones and false-branching. However, the strain KPABG 132178 does not have meristematic zones and is characterized by olive-green cell coloration in contrast with the type material, though it reveals its identity in the 16S rRNA and has the highest similarity of 16S-23S ITS rRNA genes with the type strain *Drouetiella hepatica* UHER 2000/2452. Based on the analyses of an enlarged dataset, and with the possibility of estimating the infraspecific variability with strains from Europe, the Russian Arctic, Alaska and Antarctica, we assume it is appropriate to treat the unnamed strains of *Drouetiella* (V16 and ANT.L52.2) as belonging to *Drouetiella hepatica,* taking into account a <1.3% variation in 16S rRNA genes, <7% variation in the 16S-23S ITS rRNA, and minor changes in the helices' structures; this is in opposition to the study by Mai et al. [22]; wherein, strain ANT.L52.2 is proposed as a candidate for a new species. Possibly, *Drouetiella hepatica* is a more molecularly conservative species, opposite to that of *Drouetiella lurida.*

The single tested strain of *Drouetiella fasciculata* from the USA is clearly distinct from other *Drouetiella* species due to its bright blue-green color and fasciculation of trichomes [22]. The type strain of this species shares only 96.42% identity with the *Drouetiella lurida* type strain and 96.54% identity with the *D. hepatica* type strain in the 16S rRNA, 79.57% and 80.59% in 16S-23S ITS rRNA, correspondingly; *Drouetiella fasciculata* is also distinguished by its secondary structures of D1-D1' and V3 helices. Taking into account the 16S rRNA sequence divergence between the recently described sister-related genera *Pegethrix* and *Cartusia* (98.0% ± 0.57), or *Pegethrix* and *Elainella* (97.0% ± 0) [22], *Drouetiella fasciculata* could be supposed to transfer to a distinct genus.

The new species described here, *Drouetiella ramosa,* is morphologically the most similar to *D. lurida,* though it differs mainly by its very common false-branching and shorter cells. Molecularly, *Drouetiella ramosa* is allied to the *D. hepatica* clade, from both tested nucleotide loci. The differences in the *D. ramosa* and *D. hepatica* count, 1.35–1.38% in the 16S rRNA gene and 8.43–10.76% in the 16S-23S ITS rRNA, that demonstrated more high level of differentiation than level of infraspecific variability in both *D. lurida* and *D. hepatica*. The D1-D1', Box-B, and V2 and V3 helices' structures are similar to *D. hepatica,* but are not identical in the number of nucleotide substitutions, reflected in the unique motives in the terminal loops (Figures 6–9). We suggest that *D. ramosa* fits well in concept, of delimitation of a new species according to the published and mostly accepted criteria of similarity between cyanobacterial species [44,45]. The puzzling combination of morphological and molecular features of *Drouetiella ramosa* makes its identification difficult, and can only be robustly resolved by testing the DNA sequence data.

*Drouetiella* species occur in several types of habitats. Evidently, *Drouetiella lurida* is widely distributed in different ecological conditions: moist soils (KPABG 41662, 610005) [46], stagnant water and lake littoral zones [46], epilithic on a boulder in a river (KPABG 4163), and in thermal springs [47,48]. Both of the *Drouetiella hepatica* strains occurred on a subaerial seep wall near the water. *Drouetiella fasciculata* was found in subaerophytic habitats in wet walls. *Drouetiella ramosa* grows in aerophytic habitats in biological soil crusts.

*Drouetiella lurida* was found in different regions and could be characterized as a worldwide distributed species [46]. The occurrence of *Drouetiella lurida* at high latitudes has

not been recorded previously. In Russia the species was only noted in the Caucasus [47]. The distribution of *Drouetiella hepatica,* considering the inclusion of the species' strains V16 and ANT.L52.2, comprises both polar regions. The occurrence of *Drouetiella hepatica* in the Polar Urals has been recorded for the first time for this Russian flora. Obviously, reliable Arctic and Subarctic cyanobacterial species identification remains extremely important to understand their geographic distribution.

**Author Contributions:** Conceptualization, D.D.; methodology, D.D. and A.V.; formal analysis, D.D. and A.V.; investigation, D.D., I.N., A.V. and E.P.; resources, D.D.; data curation, D.D. and A.V.; writing—original draft preparation, D.D. and A.V.; writing—review and editing, D.D., A.V., I.N. and E.P.; visualization, D.D. and A.V.; supervision, D.D.; project administration, D.D.; funding acquisition, D.D. and A.V. All authors have read and agreed to the published version of the manuscript.

**Funding:** This research was funded by the Russian Science Foundation, grant number 21-14-00029 (https://rscf.ru/project/21-14-00029/ accessed on 18 November 2022). The research was partially carried out within the framework of the state tasks Polar-Alpine Botanic Garden-Institute No. 1021071612832-8-1.6.11 (FMER-2021-0001) and IB FRC Komi SC UB RAS No. 122040600026-9. The research was performed using the large-scale research facilities of the herbarium at the Polar-Alpine Botanic Garden-Institute (KPABG; Kirovsk, Russia) reg. No. 499397.

**Institutional Review Board Statement:** Not applicable.

**Informed Consent Statement:** Not applicable.

**Data Availability Statement:** The data supporting the reported results can be found in the publicly archived dataset of the "L" Information system https://isling.org/ (accessed on 10 November 2022).

**Conflicts of Interest:** The authors declare no conflict of interest. The funders had no role in the design of the study; in the collection, analyses, or interpretation of the data; in the writing of the manuscript; or in the decision to publish the results.

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
