# Peer review of "Terrestrial Species of Drouetiella (Cyanobacteria, Oculatellaceae) from the Russian Arctic and Subarctic Regions and Description of Drouetiella ramosa sp. nov."

_diversity, doi:10.3390/d15020132_

Round 1
Reviewer 1 Report
Manuscript ID: diversity-2074880
This manuscript describes a new Drouetiella species from the Russian Arctic and Subarctic regions. The authors provided morphological features, and molecular and ecological information to support the separation of new Drouetiella species, D. ramosa. This manuscript was well-structured. The main issue of the study in my opinion is that the authors should explain and prove that Drouetiella ramosa is really separate from D. hepatica. It may be shown in results in some ways, but not discussed and pointed out strongly enough. Please add this point both to the results and to the discussion. My general comments are showed as below
- Line 42-45: I think that this information lack some Oculatellaeae genus newly described such as Siamcapillus and Shackletolyngbya. Could you please check and add more references or information?
- Line 47-55: I do not think Diversity requires authorities on names, as appears here for described Druetiella species except for new species in the taxonomic description. Furthermore, the authors did not add authorities for other cyanobacterial genera mentioned in this manuscript. Thus, I recommend authors should revise in the same format. If it is not required (no mention in the guideline) these authorities may be removed.
- Line 136: Why was the Bayesian analysis (BI) provided only the 16S rRNA gene? I recommend that BI analysis also should conduct for 16S-23S ITS data set.
- Line 144: Could you please check the number of trees sampled? I think that the value of 15000 trees may be wrong. Should it be 7500 trees? You obtained 10000 trees (1000000/100) from BI analysis and 2500 trees were discarded. Thus, it should remain 7500 trees for analysis.
Line 226 : “...relationship with BS=72.
-Line 149 “ The infrageneric and infraspecific variability of 16S rRNA and 16S-23S ITS sequences”
In this article, most authors mentioned and compared the infraspecific variability between Drouetiella species more than the infrageneric value, which was only stated in Table 3. Thus, I think that the wording of the infrageneric value may not be necessary to mention in this article.
- It is somewhat confusing because the authors mentioned the p-distance values or p-distance dissimilarity or variability (i.e. Table 3, L228, 329) in many places in the manuscript. However, the authors presented in this article only the sequence similarity percent calculated reversely from the genetic p distance value. Thus, I recommend authors consider the revision and should use the same wording, such as the sequences similarity, in the explanation of genetic variability.
Furthermore, in Table 3, the similarity values were not in the format of 16S value/ITS value (i.e. 99.90/99.12) as described in the table heading. Could authors please revise this point to easily understand readers?
- Line 256 : For the ITS data, some cyanobacterial genera/species show 1-4 operons such as Trichotorquatus or Siamcapillus. How many operons did authors find in your species? Which operon did you use for all ITS analyses? I recommend author should explain or add more information about this point.
- Line 363 : “ a high level of plasticity complicate the morphological identification of species in the genus Drouetiella.” I am not sure that morphological plasticity could be found in the Drouetiella. Thus, the authors should add references to support this.
- Line 377: I do not think that the secondary structure could allow a KPABG61005 to identify as Drouetiella lurida when considering the difference of the D1-D1’ structure, ITS sequences similarity below species separation threshold which should be >3% dissimilarity or < 97% similarity (Gonzalez-Resendiz et al. 2019), phylogenetic positions or morphological features. This is in contrast with the sentence in line 404 “the D1-D1’ showed differences among all Drouetiella species”. D1-D1’ structure of a KPABG61005 was very different compared with other Drouetiella species. Why do you think that this strain could be identified as D. lurida? Please discuss.
- Furthermore, I also checked data in the Genbank and found that the 16S-23S ITS sequences of Drouetiella lurida Lukesova 1986/6 and D. hepatica Uher 2000/2452 have only one operon type containing tRNA regions. Which operon did you use for analyses? Why did not you analyze the V2 helix locating between two tRNA genes if you used the same operon of the type strains containing tRNA regions? Could you please show an alignment of the ITS regions used in this study as the supplemental material?
- As explained in the manuscript, it is difficult to identify Drouetiell at the species level by morphological features. However, in this study, it also looks that a strain SYKOA C-013-09 could be classified as D. hepatica considering its position in 16S rRNA and 16S-23S ITS phylogenic trees. When investigated roughly, the D1-D1’, Box-B and V3 helices structures of a strain SYKOA C-013-09 were very similar to those of D. hepatica Uher 2000/2452, the reference strain, but different the type of nucleotide in some position. Thus, I think that the authors should add more discussion or provide more information to indicate that Drouetiella ramosa is really separate from D. hepatica.
Author Response
We appreciate your time and productive suggestions which definitely improved Drouetiella manuscript. We agree with most corrections, and we included them into the new version of the article. Our changes represented in the text below:
- Line 42-45: I think that this information lack some Oculatellaeae genus newly described such as Siamcapillus and Shackletolyngbya. Could you please check and add more references or information?
Changed, thanks
- Line 47-55: I do not think Diversity requires authorities on names, as appears here for described Druetiella species except for new species in the taxonomic description. Furthermore, the authors did not add authorities for other cyanobacterial genera mentioned in this manuscript. Thus, I recommend authors should revise in the same format. If it is not required (no mention in the guideline) these authorities may be removed.
Fixed. We deleted authorities of genera, but still use them in species.
- Line 136: Why was the Bayesian analysis (BI) provided only the 16S rRNA gene? I recommend that BI analysis also should conduct for 16S-23S ITS data set.
We have no possibility to produce the ITS dataset similar to 16S rRNA gene dataset in course of accession sampling due to absence of ITS sequences for majority of accessions. Obtained ITS dataset for 11 accessions is quite enough to be analyzed only from ML procedure.
- Line 144: Could you please check the number of trees sampled? I think that the value of 15000 trees may be wrong. Should it be 7500 trees? You obtained 10000 trees (1000000/100) from BI analysis and 2500 trees were discarded. Thus, it should remain 7500 trees for analysis.
We obtained 10000 trees in each run, the first 2500 trees in each run were discarded as burn-in. 7500 trees stay in each run, but 15000 trees were sampled from both runs. It is correct.
- Line 226 : “...relationship with BS=72.
Сhanged on: ...supported by BS=72%
- Line 149 “ The infrageneric and infraspecific variability of 16S rRNA and 16S-23S ITS sequences” In this article, most authors mentioned and compared the infraspecific variability between Drouetiella species more than the infrageneric value, which was only stated in Table 3. Thus, I think that the wording of the infrageneric value may not be necessary to mention in this article.
We suppose that term “infraspecific variability” suggested variability among different strains of single species, the term “infrageneric variability” suggested variability among different species of single genus (as well could be used the term “interspecific variability”. But we could not understand what means phrase “infraspecific variability between Drouetiella species”. In this study we discussed “infraspecific variability” and “infrageneric (interspecific) variability”.
- It is somewhat confusing because the authors mentioned the p-distance values or p-distance dissimilarity or variability (i.e. Table 3, L228, 329) in many places in the manuscript. However, the authors presented in this article only the sequence similarity percent calculated reversely from the genetic p distance value. Thus, I recommend authors consider the revision and should use the same wording, such as the sequences similarity, in the explanation of genetic variability.
Changed, thanks
- Furthermore, in Table 3, the similarity values were not in the format of 16S value/ITS value (i.e. 99.90/99.12) as described in the table heading. Could authors please revise this point to easily understand readers?
Changed, we added /, thanks
- Line 256 : For the ITS data, some cyanobacterial genera/species show 1-4 operons such as Trichotorquatus or Siamcapillus. How many operons did authors find in your species? Which operon did you use for all ITS analyses? I recommend author should explain or add more information about this point.
The single operon of ITS was obtained for each of five Droutiella strains. This information is added in 2.4.
- Line 363 : “ a high level of plasticity complicate the morphological identification of species in the genus Drouetiella.” I am not sure that morphological plasticity could be found in the Drouetiella. Thus, the authors should add references to support this.
It is a mistake. Of course, we are mean "a low level of plasticity".
- Line 377: I do not think that the secondary structure could allow a KPABG61005 to identify as Drouetiella lurida when considering the difference of the D1-D1’ structure, ITS sequences similarity below species separation threshold which should be >3% dissimilarity or < 97% similarity (Gonzalez-Resendiz et al. 2019), phylogenetic positions or morphological features. This is in contrast with the sentence in line 404 “the D1-D1’ showed differences among all Drouetiella species”. D1-D1’ structure of a KPABG61005 was very different compared with other Drouetiella species. Why do you think that this strain could be identified as D. lurida? Please discuss.
Gonzalez-Resendiz et al. (2019) pointed that 3-7% differences in ITS is questionable to limit a distinct species: the “recognizable discontinuity” should be between two units. In our case the intermediate position between type D. lurida and KPABG61005 belongs to strain KPABG 41662. Thus, we should consider a three taxa - D. lurida Lukesova 1986/6+D. lurida KPABG 4163, strain KPABG 41662, strain KPABG61005 – or except a single one. Our experience with multiplied sampled species of the genus Phormidesmis (Davydov, Vilnet, 2022) revealed in ITS infraspecific variation till 7.05% in P. nigrescens (24 strains), 7.62 in P. priestley (15 strains), 6.68% P. communis (19 strains). Moreover, P. nigrescens and P. priestley by ITS composed an intermingled clade and dissimilarity between them by ITS a little bit higher than infraspecific - 8.99! Each of three mentioned species obtained more than one secondary structures in D1 and BoxB. There is a question: should we describe a number of undifferentiated morphologically species based on secondary structures differences within known monophyletic units with recognizable morphology or should accept species variability in ITS region? We suppose that limited sampling (number of accessions and geographical localities) resulted in numerous described species that possibly could be result only infraspecific variability of known species. From the other hand Yarza et al. (2014) pointed differences >98.7% in 16S between strains as enough to treat them as distinct species. Similarity among all tested strains of D. lurida counts 99.17-99.90% (Table 3). Thus, the splitting the D. lurida we think prematurely based on current data.
- Furthermore, I also checked data in the Genbank and found that the 16S-23S ITS sequences of Drouetiella lurida Lukesova 1986/6 and D. hepatica Uher 2000/2452 have only one operon type containing tRNA regions. Which operon did you use for analyses? Why did not you analyze the V2 helix locating between two tRNA genes if you used the same operon of the type strains containing tRNA regions? Could you please show an alignment of the ITS regions used in this study as the supplemental material?
We use the same operon. Information concerning to V2 secondary structures and figure are added in the manuscript. If it is necessary according with rules we could provived an alignment, but each reader could produce it from GenBank accession, data are open since Davydov, Vilnet, 2022 was published.
- As explained in the manuscript, it is difficult to identify Drouetiell at the species level by morphological features. However, in this study, it also looks that a strain SYKOA C-013-09 could be classified as D. hepatica considering its position in 16S rRNA and 16S-23S ITS phylogenic trees. When investigated roughly, the D1-D1’, Box-B and V3 helices structures of a strain SYKOA C-013-09 were very similar to those of D. hepatica Uher 2000/2452, the reference strain, but different the type of nucleotide in some position. Thus, I think that the authors should add more discussion or provide more information to indicate that Drouetiella ramosa is really separate from D. hepatica.
The similarity of newly described D. ramosa from D. hepatica counts 98.65-98.92% in 16S and 89.24-91.57% in ITS that higher than level of infraspecific variability in both known species – D. lurida and D. hepatica. We suggested D. ramosa as a new species according with published criteria of similarity between cyanobacterial species (Gonzalez-Resendiz et al., 2019, Yarza et al. 2014)). The D1-D1’, Box-B and V3 helices structures are similar to D. hepatica but not identical in course of nucleotide substitutions.
Reviewer 2 Report
This study used the polyphasic approach to characterize five strains of Drouetiella isolated from Russian Arctic and Subarctic area and created a new species as D. ramosa sp. nov. according to the International Code of Nomenclature for Algae, Fungi, and Plants. The methodological approach has been correctly used. Basically, I agree the establishment of new species. The paper can be accepted after following minor revisions:
1. Drouetiella ramosa sp. nov. : sp. nov. in title and in the text should not be italic
2. filament or cellular color of Drouetiella ramosa was descripted as green or olive green, but it was also described as colony browlish in line 343, which is not consistent
3. C in Fig.2 and b,d in Fig 3 showed true branch somehow, rather than did false-branch exactly or clearly. C in Fig.3 was exactly false-branched
Author Response
We appreciate your time and productive suggestions which definitely improved Drouetiella manuscript.
- Drouetiella ramosa sp. nov.: sp. nov. in title and in the text should not be italic
Changed, thanks.
- filament or cellular color of Drouetiella ramosa was descripted as green or olive green, but it was also described as colony browlish in line 343, which is not consistent
Changed, thanks.
- C in Fig.2 and b,d in Fig 3 showed true branch somehow, rather than did false-branch exactly or clearly. C in Fig.3 was exactly false-branched
We appreciate the note but couldn't agree with this, because all of the filaments have a obvious false-branching in pictures.
Reviewer 3 Report
This is an interesting paper with a grounded morphological, molecular and ecological data. Congratulations to the authors for the well-done work.
I have only one optional recommendation: despite English diagnosis is acceptable, it is always better to support with a Latin one.
Author Response
We appreciate your time and productive suggestions which definitely improved Drouetiella manuscript.
Round 2
Reviewer 1 Report
Thank you very much for your response to all comments. I am Ok with your revised manuscript. Now, this new version is appropriate for publication.